# ActiveAD: Planning-Oriented Active Learning for End-to-End Autonomous Driving

## Abstract

End-to-end differentiable learning has emerged as a prominent paradigm in autonomous driving (AD). A significant bottleneck in this approach is its substantial demand for high-quality labeled data, such as 3D bounding boxes and semantic segmentation, which are especially expensive to annotate manually. This challenge is exacerbated by the long tailed distribution in AD datasets, where a substantial portion of the collected data might be trivial (*e.g.* simply driving straight on a straight road) and only a minority of instances are critical to safety. In this paper, we propose ActiveAD, a planning-oriented active learning strategy designed to enhance sampling and labeling efficiency in end-to-end autonomous driving. ActiveAD progressively annotates parts of collected raw data based on our newly developed metrics. We design innovative diversity metrics to enhance initial sample selection, addressing the cold-start problem. Furthermore, we develop uncertainty metrics to select valuable samples for the ultimate purpose of route planning during subsequent batch selection. Empirical results demonstrate that our approach significantly surpasses traditional active learning methods. Remarkably, our method achieves comparable results to state-of-the-art end-to-end AD methods - by using only 30% data in both open-loop nuScenes and closed-loop CARLA evaluation.

## 1 Introduction

Autonomous driving (AD), as one of the most exciting applications of AI, has drawn increasing attention. Traditional AD systems are usually module-based which divide the driving task into sub-tasks: perception (Li et al., 2022; Huang et al., 2021; Liu et al., 2023), prediction (Shi et al., 2022; Jia et al., 2022b;a; 2023b), planning (Treiber et al., 2000; Dauner et al., 2023), etc. However, modular systems suffer from error accumulations, less principled optimization, and redundant computations due to the separate training objectives of each sub-task, which limit the performance upper bound (Chen et al., 2023). On the other hand, the success of LLM (Brown et al., 2020; OpenAI, 2023) has demonstrated the power of the data-driven scalable paradigm (Wu et al., 2023; Yang et al., 2023). Motivated by these insights, the shift towards end-to-end AD (E2E-AD) has recently emerged as a promising area (Hu et al., 2023). These latest works take advantages of data-driven approaches, as well as mitigate the limitations of modular frameworks.

A major factor behind the success of LLM is the abundance of almost free text data available online. This is not the case in autonomous driving (AD), where state-of-the-art E2E-AD systems such as UniAD (Hu et al., 2023) and VAD (Jiang et al., 2023) are still confined by supervised learning. It requires fine-grained annotations including 3D bounding boxes of agents and semantic segmentation for lanes and traffic signs, which are quite expensive. Therefore, labeling becomes one significant bottleneck of the scaling up process of these end-to-end methods. Even worse, it is widely acknowledged that the AD task has serious long-tailed issues (Jain et al., 2021), which means a large part of collected data is trivial *e.g.* simply driving forward in a straight road, and only a few cases are safety-critical. Such imbalances in data annotation further limit the application of data-driven methods and significantly increase the cost of ineffective annotation. Thus, strategies to alleviate data-related issues are of prime importance.

To address these issues, we first pose a fundamental question: *Is it necessary to annotate all collected raw data to achieve optimal performance*? Through empirical studies, we demonstrate that the answer is *NO*. Further, we explore the way to select the most useful samples to annotate for training, which

Figure 1: **Active Learning scheme for End-to-End Autonomous Driving.** We formulate a comprehensive pipeline and meticulously design a task-specific active selection strategy for choosing initial samples as well as incremental samples in subsequent iterations.

belongs to the active learning task (Zhan et al., 2022). Different from existing literature focusing on the perception part (Luo et al., 2023b), inspired by the planning-oriented philosophy in UniAD (Hu et al., 2023), we design an active learning method called ActiveAD, which leverages planning routes and scores to directly optimize planning.

There are several major gaps in adopting existing active learning methods (Gal et al., 2017; Kirsch et al., 2019; Ash et al., 2019; Yoo & Kweon, 2019; Sinha et al., 2019) to AD. Firstly, data in AD often involves rich multi-modality information, such as video streams, driving trajectories, and various meta-information like vehicle speed, whereas most existing active learning methods typically consider only single-modal images as input. Secondly, AD tasks can be more complex than simple classification, yet many existing methods are confined to this paradigm (Gal et al., 2017; Kirsch et al., 2019; Ash et al., 2019). Therefore, it calls for adaption to better handle the diverse inputs and optimization targets in AD.

Fig. 1 illustrates the designed scheme of the active learning paradigm for end-to-end autonomous driving (AD), addressing the identified challenges and enhancing the utilization of task-relevant information. In the initial sample selection stage, ActiveAD introduces Ego-Diversity, replacing the commonly used random selection in traditional active learning paradigms (Sener & Savarese, 2018; Sinha et al., 2019). Ego-Diversity effectively leverages nearly free information within raw AD data, considering factors such as weather, lighting, and vehicle speed. During the iterative process of active sample selection, we propose three intuitive and effective metrics: Displacement Error, Soft Collision, and Agent Uncertainty. Displacement Error utilizes the recorded ego trajectory as a concise yet essential metric. Soft Collision calculates the potential for collisions based on the predicted trajectory of the ego vehicle and the trajectories of other objects, serving as a continuous version of the collision rate. Agent Uncertainty assesses the uncertainty of other vehicles in complex road conditions.

Extensive experiments are conducted to validate the proposed ActiveAD. ActiveAD significantly outperforms general active learning methods. Under a 30% annotation budget, ActiveAD achieves comparable or even slightly better planning performance than state-of-the-art methods trained on the complete dataset. In the ablation study, we provide a detailed analysis of the contribution and effectiveness of the designed metrics, examining the robustness of performance across different scenarios. Additionally, we provide visualizations and analyses of the results for different selection choices. Our contributions can be summarized as follows:

- To the best of our knowledge, **we are the first to delve into the data problems and address the challenges in end-to-end autonomous driving (E2E-AD).** We provide a simple yet effective solution to identify and annotate valuable data for planning within a limited budget.
- Based on the planning-oriented philosophy of end-to-end methods, we design a **novel task-specific diversity and uncertainty measurement** for the planning routes.
- Extensive experiments and ablation studies demonstrate the effectiveness of our approach. **Using only** 30% **training data, ActiveAD outperforms general peer methods by a large margin and achieves comparable performance to the SOTA method training with the entire** 100% **dataset in both open-loop nuScenes and closed-loop CARLA evaluation.**

## 2 RELATED WORK

### 2.1 END-TO-END AUTONOMOUS DRIVING

The concept of end-to-end autonomous driving has roots dating back to the 1980s (Pomerleau, 1988). In the era of deep learning, early efforts focused on the straightforward mapping (Muller et al., 2005). Subsequently, (Zhang et al., 2021; Li et al., 2024) explored the application of reinforcement learning to develop an end-to-end driving policy. Some state-of-the-art student models (Wu et al., 2022; Hu et al., 2022a) are developed based on them while PlanT (Renz et al., 2022) suggested employing a Transformer for the teacher model. LBC (Chen et al., 2020) and DriveAdapter (Jia et al., 2023a) involved initially training a teacher model with privileged inputs. In later works, multiple sensors are used. Transfuser (Prakash et al., 2021; Chitta et al., 2022) employed a Transformer for camera and LiDAR fusion. LAV (Chen & Krähenbühl, 2022) adopted PointPainting (Vora et al., 2020). Interfuser (Shao et al., 2022) injected safety-enhanced rules during the decision-making process. ThinkTwice (Jia et al., 2023c) introduced a DETR-like scalable decoder paradigm for the student model. ReasonNet proposed specific modules for student models to better exploit temporal and global information. In (Jaeger et al., 2023), they suggested formulating the output of the student as classification problems to avoid averaging. ST-P3 (Hu et al., 2022c) unified the detection, prediction, and planning tasks into the form of BEV segmentation. UniAD (Hu et al., 2023) adopted Transformer to connect different tasks. Further, VAD (Jiang et al., 2023) reduced some potential redundant modules in UniAD while demonstrating better performance.

### 2.2 ACTIVE LEARNING

Active learning algorithms exploit the limited annotation budget by selecting the most informative samples for labeling. They select data samples based on the criterion of either uncertainty or diversity. Uncertainty-based algorithms prefer those difficult samples most confusing for the models. The difficulty of each data sample may be measured by prediction entropy (Joshi et al., 2009; Luo et al., 2013), prediction inconsistency (Gao et al., 2020), loss estimation (Yoo & Kweon, 2019) or its potential influence for model training (Freytag et al., 2014; Liu et al., 2021). Alternatively, other methods pay attention to the diversity of the selected subset. Some early works (Sener & Savarese, 2018; Sinha et al., 2019) mainly consider the representation diversity in the global image level, while following papers (Agarwal et al., 2020; Liang et al., 2022) dig into the regional information to deal with fine-grained detection or segmentation tasks. Furthermore, some recent works (Xie et al., 2023a;b; Yi et al., 2022) utilize the strong representation ability of models pretrained on large datasets to measure the image diversity of the target dataset more accurately. Recently, CRB (Luo et al., 2023b) has pioneered active learning to LiDAR-based 3D object detection and KECOR (Luo et al., 2023a) greedily select informative point clouds by maximizing the kernel coding rate in AD.

However, most prior works focus on the traditional tasks like classification, detection, or segmentation, but the recently prominent planning-oriented end-to-end AD setting is hardly explored. Instead of just simple prediction probability, The task model outputs the future ego-vehicle trajectory. Besides, this task requires to reason from the interaction (Jia et al., 2021b) between ego-vehicle and surroundings, which cannot be reflected from superficial visual patterns. To this end, we fill in this gap by devising novel uncertainty and diversity metrics for active learning of end-to-end AD.

## 3 FORMULATION OF ACTIVE LEARNING FOR END-TO-END AD

State-of-the-art end-to-end autonomous driving (AD) methods (Hu et al., 2022b; Jiang et al., 2023) typically take raw sensor data as inputs and generate planned trajectories for the ego vehicle. To facilitate training and mitigate overfitting, additional annotations such as 3D bounding boxes of agents and semantic segmentation of lanes (Caesar et al., 2020) are used. Since the collected raw data is usually in the form of clips containing multiple temporal frames of surrounding images and canbus information, organizing annotations at the clip level offers several benefits. Firstly, it streamlines the annotation process by providing a coherent context for labeling. Secondly, it enables the establishment of spatiotemporal connections between objects. Therefore, we choose to treat each clip as a distinct unit rather than considering individual frames. This approach is also consistent with current practices in AD research (Caesar et al., 2020).

Formally, we define the active learning task for end-to-end AD as follows: denote $\mathcal{I}^t$ as the raw sensor data in the frame $t$ where $t \in [T] = \{1, 2, ..., T\}$ and $T$ is the length of its corresponding clip $\mathcal{S}_i$. Apart from the raw sensor data, the recorded trajectory $\tau_i$ and states $e_i$ (speed $v_i$ and driving commands $cmd_i$) of the ego vehicle, weather condition $w_i$ (Sunny or Rainy) and the lighting condition $l_i$ (Day or Night) are also annotation-free or extremely cheap to obtain. For simplicity, we denote these easy-to-obtain labels as $\mathcal{O}_i = (e_i, w_i, l_i)$. For the scene that has not been meticulously annotated (e.g., without annotations of 3D bounding boxes and semantic segmentation), we can represent such information as $X_i = (\mathcal{S}_i, \tau_i, \mathcal{O}_i)$ where $i \in [N]$ and $N$ is the number of scenes.

For the labels that require meticulous annotation, we denote them as $Y_i$. $Y_i = (\mathcal{A}_i, \mathcal{B}_i, \mathcal{C}_i)$ where $\mathcal{A}_i$ donates attributes (visibility, activity, and pose), $\mathcal{B}_i$ denotes the 3D bounding box and $\mathcal{C}_i$ donates the semantic segmentation of lanes (Caesar et al., 2020).

Initially, we have the access to the unlabeled data pool $\mathcal{P}^u = \{X_i\}_{i \in [N]}$. Under the given annotation budget $B$ where $|B| < N$, one should select the index set $\mathcal{K} = \{k_i \in [N]\}_{i \in [B]}$ to obtain the subset $\mathcal{P}^u_{\mathcal{K}} = \{X_{k_i}\}_{i \in [B]} \subset \mathcal{P}^u$ from $\mathcal{P}^u$ and acquire the related labels $\{Y_{k_i}\}_{i \in [B]}$. Then the models are trained on the labeled set $\mathcal{P}^l_{\mathcal{K}} = \{(X_{k_i}, Y_{k_i})\}_{i \in [B]}$. The objective is to choose the sampling strategy to select the labeled set under the budget to minimize the expectation error of the model, which usually refers to the L2 loss and collision ratio (Hu et al., 2022b; Jiang et al., 2023) in end-to-end AD.

The active selection process involves the following steps: 1) Select a subset of data as the initial set. 2) Train a model based on the current data. 3) Utilize the trained model's features and outputs to select a new subset of data based on a designed strategy. 4) Repeat steps 2 and 3 until the budget is reached. Fig. 1 demonstrates the pipeline of our method and the details process is shown in Sec. 4.3.

## 4 ACTIVEAD METHOD

We provide a detailed description of our method ActiveAD, within the framework of end-to-end autonomous driving (AD). Leveraging the characteristics of data specific to AD, we devise corresponding metrics for diversity and uncertainty. Sec. 4.1 introduces the methodology for designing diversity metrics, which are utilized as criteria for selecting the initial set. Sec. 4.2 presents the design of uncertainty metrics to identify more challenging data samples. Sec. 4.3 summarizes the entire ActiveAD process and provides a detailed algorithmic depiction.

### 4.1 INITIAL SAMPLE SELECTION FOR LABELING

For active learning in computer vision, the initial sample selection is often solely based on raw images without extra information or learned features, leading to the common practice of **Random** initialization (Sener & Savarese, 2018; Sinha et al., 2019; Yoo & Kweon, 2019; Kim et al., 2021; Parvaneh et al., 2022). For AD, there is additional prior information to leverage. Specifically, when collecting data from sensors, conventional information such as the speed and trajectory of the ego vehicle can be simultaneously recorded. Additionally, weather and lighting conditions are generally continuous and easy to annotate in the clip-level. The information can benefit making choices for the initial set selection. Therefore, we design the **Ego-Diversity** metric for initial selection.

**Ego-Diversity** consists of three components: 1) weather-lighting 2) driving commands 3) average speed. Inspired by the setting in (Liu et al., 2023; Zhu et al., 2023), we firstly divide the complete dataset into four mutually exclusive subsets: Day-Sunny (DS), Day-Rainy (DR), Night-Sunny (NS), Night-Rainy (NR), using the description in nuScenes (Caesar et al., 2020). Secondly, We categorize each subset based on the number of left, right, and straight driving commands (Hu et al., 2022c;b; Jiang et al., 2023) within a complete clip into four categories: Turn Left (L), Turn Right (R), Overtake (O), Go Straight (S). We design a threshold $\tau_c$, where if the numbers of left and right commands in a clip are both greater than or equal to the threshold $\tau_c$, we consider it as an overtaking behavior in this clip. If only the number of left commands is greater than the threshold $\tau_c$, it indicates a left turn. If only the number of right commands is greater than the threshold $\tau_c$, it indicates a right turn. All the other cases are considered as going straight. Thirdly, we calculate the average speed in each scene and sort them in ascending order in the related subset.

Given the initial annotation budget $n_0$, we should split the numbers to each subset. We define the original number of each subset $s$ as $n_s$ and the selected number to label as $n_s^l$. The number of samples

in different categories often varies, and samples from minority categories (such as Night-Rainy and Overtake) are typically challenging and critical, requiring more attention. Therefore, we introduce a parameter $\gamma$ to control the proportions of each subset $P_s$. The proportion calculation of first-level weather-lighting subset is specified as follows:

$$P_x = \frac{n_x^{\gamma}}{\sum_{z \in \{\text{DS, DR, NS, NR}\}} n_z^{\gamma}}, \text{ where } x \in \{\text{DS, DR, NS, NR}\}. \tag{1}$$

The number of annotations for each subset $s$ is $n_s^l = n_0 P_s$. By decreasing $\gamma$, we can increase focus on minority classes. When $\gamma = 1$, it indicates an absolute uniform distribution, where each category is chosen equally. If $\gamma < 1$, it signifies a bias towards categories with fewer total samples. For the second-level subset consisting of four driving scenarios, the process is similar as follows:

$$P_{x,y} = P_x \times \frac{n_{x,y}^{\gamma}}{\sum_{z \in \{\text{L, R, O, S}\}} n_{x,z}^{\gamma}}, \text{ where } x \in \{\text{DS, DR, NS, NR}\} \text{ and } y \in \{\text{L, R, O, S}\}. \tag{2}$$

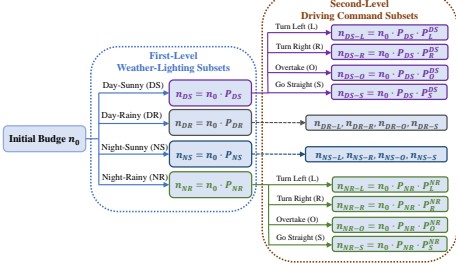

Fig. 2 illustrates the detailed and intuitive selection process for the initial sample based on a multi-way tree. First, the entire dataset is divided into four first-level subsets: DS, DR, NS, and NR, according to weather and lighting conditions. Second, within each of these subsets, further divisions are made based on driving commands, resulting in four second-level subsets: L, R, O, and S, from each weather-lighting subset. Finally, based on the available sample budget $n_{x,y} = n_0 P_{x,y}$ in each second-level subset, selections are made at regular intervals within the sorted speeds.

Figure 2: **Ego-Diversity Initialization.**

### 4.2 CRITERION DESIGN FOR INCREMENTAL SELECTION

In this section, we describe how we incrementally annotate new clips based on a model trained with previously annotated clips. We use the intermediate model to infer on the unannotated clips, and the subsequent selection is based on these outputs. From a planning-oriented perspective, we introduce three criteria for data selection: Displacement Error, Soft Collision, and Agent Uncertainty.

**Criterion I: Displacement Error (DE).** Denote $\mathcal{L}_{DE}$ as the distance between the predicted planning route $\tau$ of the model and the human trajectory $\tau^*$ recorded in the dataset.

$$\mathcal{L}_{DE} = \frac{1}{T} \sum_{t=1}^{T} \|\tau_t - \tau_t^*\|_2, \tag{3}$$

where $T$ represents the frames in the scenes. Since Displacement Error is a performance metric that does not require annotation which is inherently recorded during the data collection process in autonomous driving, it naturally becomes the first and most crucial criterion in active selection.

**Criterion II: Soft Collision (SC).** Define $\mathcal{L}_{SC}$ as the distance between the predicted ego-trajectory and predicted agent-trajectory. Similar to (Jiang et al., 2023), we will filter out low-confidence agent predictions by a threshold $\epsilon_a$. In each scene, we select the shortest distance as a measure of the danger coefficient. Simultaneously, we maintain a positive correlation between the term and the closest distance:

$$\mathcal{L}_{SC} = \sum_{t=1}^{T} \exp\left(-\min_{a \in \text{agents}} (\tau_{t,ego} - \tau_{t,a})\right). \tag{4}$$

We use Soft Collision as one of the criteria for several reasons. Firstly, unlike Displacement Error, calculating Collision Ratio depends on the annotation of 3D bounding boxes for objects, which are not available in unlabeled data. Therefore, we need a criterion that can be calculated solely based on the model's inference results. Secondly, using a Hard Collision criterion—where a predicted ego trajectory collides with other predicted agents' trajectories (assigned as 1 for collision and 0 otherwise)—could result in too few positive samples, as the collision rate of state-of-the-art models

in AD is usually very low (less than 1%). Thus, we use the closest distance from the ego vehicle to other objects as a substitute for the Collision Rate metric. When the distance to other vehicles or pedestrians is too close, the risk of collision is significantly higher. In summary, Soft Collision serves as an effective indicator to measure the likelihood of a collision, providing dense supervision.

**Criterion III: Agent Uncertainty (AU).** The prediction of surrounding agents' future trajectories naturally has uncertainty (Jia et al., 2021a) and thus the motion prediction module usually generates multiple modalities and corresponding confidence scores. We aim to select those data where nearby agents has high uncertainties. Specifically, we filter out faraway agents by a distance threshold $\delta_d$ and calculate the weighted entropy of the predict probabilities of multiple modalities of remaining agents. Suppose the number of the modalities is $N_m$ and the confidence scores of a agent under different modalities are $\mathcal{P}_i(a)$ where $i \in \{1, ..., N_m\}$. Then, the Agent Uncertainty $\mathcal{L}_{AU}$ can be defined as :

$$\mathcal{L}_{AU} = \sum_{a \in \text{agent}} \mathcal{W}(a)\mathcal{H}(a) = -\sum_{a \in \text{agent}} \exp(\delta_d - d_a) \left( \sum_{i=1}^{N_m} \mathcal{P}_i(a) \log \mathcal{P}_i(a) \right), \quad (5)$$

where $\delta_d$ is the distance threshold, $d_a$ is the predicted distance between the agent and the ego vehicle, $\mathcal{W}$ represents the weight, and $\mathcal{H}$ represents the entropy.

**Overall Loss.** The loss of the samples in active selection is defined as:

$$\mathcal{L} = \mathcal{L}_{DE} + \alpha\mathcal{L}_{SC} + \beta\mathcal{L}_{AU}, \quad (6)$$

where $\alpha, \beta$ are hyper-parameters. We select the top $n_i$ unannotated clips with the largest overall loss, where $n_i$ denotes the number of clips that can be annotated in iteration $i$.

### 4.3 OVERALL ACTIVE LEARNING PARADIGM

In summary, Alg. 1 presents the entire workflow of our method. Given the available budget $B$, the initial selection size $n_0$, the number of active selections made at each step $n_i$, and $M$ total selection stages, we start by initializing the selection using randomization or the Ego-Diversity method described in Sec. 4.1. Next, we train the network using the currently annotated data. Based on the trained network, we make predictions on the unlabeled pool and calculate the overall loss as described in Sec. 4.2. Finally, we sort the samples based on the overall loss and select the top $n_i$ samples to be annotated in the current iteration. This process is repeated until the iterations reach the upper bound $M$ and the annotated number reaches the upper limit $B$.

### 5 EXPERIMENTS

We conduct experiments on the widely used nuScenes dataset (Caesar et al., 2020) in line with the peer works (Hu et al., 2023). All experiments are implemented using PyTorch and run on RTX 3090 (24G) and A100 GPUs (40G). Source code will be made publicly available.

---

**Algorithm 1 Pseudo-code for ActiveAD**

**Input:** Unlabeled pool $\mathcal{P}^u = \{X_i\}_{i \in [N]}$, labeled pool $\mathcal{P}^l = \emptyset$, model $f(\cdot; w)$, annotation budget $B$.
**Parameter**: Initial number $n_0$, active selection iterations $M$, selection number per iteration $n_{itr}$, original numbers of each subset $n_x$ and $n_{x,y}$, hyper-parameters $\alpha, \beta, \gamma$.

Initialize annotation dataset indices $\mathcal{K} = \emptyset$.
**if** *Using Ego-Diversity based initialization* **then**
  **for** *First-level subset x in {DS, DR, NS, NR}* **do**
    Calculate first-level proportion $P_x = n_x^\gamma / \sum_z n_z^\gamma$ where $z \in \{DS, DR, NS, NR\}$ by Eq. 1.
    **for** *Second-level subset y in {L, R, O, S}* **do**
      Calculate second-level proportion $P_{x,y} = P_x \times n_{x,y}^\gamma / \sum_z n_{x,z}^\gamma$ where $z \in \{L, R, O, S\}$ by Eq. 2.
      Set the annotation number $n_{x,y}^l = n_0 P_{x,y}$.
      Sort the subset $x, y$ according to the speed in ascending order and select $n_{x,y}^l$ indices at regular intervals, then add them to $\mathcal{K}$.
**else**
  Randomly select $n_0$ samples $\mathcal{K} = \{k_i \in [N]\}_{i \in [n_0]}$.
**for** $itr \in \{1, 2, \ldots, M\}$ **do**
  Update $\mathcal{P}_\mathcal{K}^u = \mathcal{P}^u - \{X_{k_i}\}_{k_i \in \mathcal{K}}$, $\mathcal{P}_\mathcal{K}^l = \{X_{k_i}, Y_{k_i}\}_{k_i \in \mathcal{K}}$.
  Train the model $f(\cdot; w)$ from scratch using $\mathcal{P}_\mathcal{K}^l$.
  Inference on unlabeled pool $\mathcal{P}_\mathcal{K}^u$ to calculate Loss $\mathcal{L} = \mathcal{L}_{DE} + \alpha\mathcal{L}_{SC} + \beta\mathcal{L}_{AU}$ in Eq. 6 for each sample.
  Sort the samples in the descending order of $\mathcal{L}$.
  Select the first $n_{itr}$ indices of the samples and add them to $\mathcal{K}$ so that $\mathcal{K} = \{k_i \in [N]\}_{i \in [\sum_{j=0}^{itr} n_j]}$.
**Output:** Return annotation indices $\mathcal{K} = \{k_i \in [N]\}_{i \in [B]}$

---

### 5.1 EXPERIMENTAL SETUP

**Dataset & Metrics.** The nuScenes (Caesar et al., 2020) dataset consists of 1,000 scenes, each lasting 20 seconds. It provides comprehensive annotations, including 3D bounding boxes for 23 classes and 8 attributes. The scenes are captured by 6 cameras, providing a 360 degree horizontal FOV, and the keyframes are annotated at a frequency of 2Hz. It covers a wide range of locations, time, and

Table 1: **Planning Performance.** ActiveAD outperforms general active learning baselines in all annotation budget settings. Moreover, ActiveAD with 30% data achieves even slightly better planning performance than using the entire dataset for training. VAD with * indicates that we have updated the results, which are better than those reported in the original works. UniAD with † indicates that we have employed the metrics from VAD to update the results (Refer to Appendix B.2 for more details).

| Base Model | Percent | Selection Method | Average L2 (m) ↓ | | | | Average Collision (%) ↓ | | | |
|---|---|---|---|---|---|---|---|---|---|---|
| | | | 1s | 2s | 3s | Avg. | 1s | 2s | 3s | Avg. |
| ST-P3 (Hu et al., 2022c) | 100% | - | 1.33 | 2.11 | 2.90 | 2.11 | 0.23 | 0.62 | 1.27 | 0.71 |
| UniAD† (Hu et al., 2023) | 100% | - | 0.42 | 0.64 | 0.91 | 0.67 | - | - | - | - |
| VAD-Base* (Jiang et al., 2023) | 100% | - | 0.39 | 0.66 | 1.01 | 0.69 | 0.08 | 0.16 | 0.37 | 0.20 |
| VAD-Tiny* (Jiang et al., 2023) | 100% | - | 0.38 | 0.68 | 1.04 | 0.70 | 0.15 | 0.22 | 0.39 | 0.25 |
| VAD-Tiny | 10% | Random | 0.51 | 0.83 | 1.23 | 0.86 | 0.40 | 0.62 | 0.98 | 0.67 |
| | 10% | ActiveFT (Xie et al., 2023b) | 0.54 | 0.88 | 1.29 | 0.90 | 0.20 | 0.41 | 0.81 | 0.47 |
| | 10% | **ActiveAD(Ours)** | **0.47** | **0.80** | **1.21** | **0.83** | **0.13** | **0.35** | **0.80** | **0.43** |
| VAD-Tiny | 20% | Random | 0.49 | 0.80 | 1.17 | 0.82 | 0.36 | 0.49 | 0.77 | 0.54 |
| | 20% | Coreset (Sener & Savarese, 2018) | 0.48 | 0.78 | 1.16 | 0.81 | 0.20 | 0.40 | 0.69 | 0.43 |
| | 20% | VAAL (Sinha et al., 2019) | 0.54 | 0.89 | 1.31 | 0.91 | 0.17 | 0.38 | 0.66 | 0.40 |
| | 20% | KECOR (Luo et al., 2023a) | 0.47 | 0.82 | 1.23 | 0.84 | 0.23 | 0.41 | 0.69 | 0.44 |
| | 20% | ActiveFT (Xie et al., 2023b) | 0.50 | 0.82 | 1.21 | 0.84 | 0.27 | 0.42 | 0.63 | 0.44 |
| | 20% | **ActiveAD(Ours)** | **0.44** | **0.73** | **1.10** | **0.76** | **0.18** | **0.36** | **0.62** | **0.39** |
| VAD-Tiny | 30% | Random | 0.45 | 0.76 | 1.12 | 0.78 | 0.17 | 0.30 | 0.63 | 0.37 |
| | 30% | Coreset (Sener & Savarese, 2018) | 0.43 | 0.71 | 1.06 | 0.73 | 0.43 | 0.51 | 0.68 | 0.54 |
| | 30% | VAAL (Sinha et al., 2019) | 0.46 | 0.79 | 1.19 | 0.81 | 0.18 | 0.33 | 0.54 | 0.35 |
| | 30% | KECOR (Luo et al., 2023a) | 0.46 | 0.78 | 1.22 | 0.82 | 0.22 | 0.43 | 0.70 | 0.45 |
| | 30% | ActiveFT (Xie et al., 2023b) | 0.46 | 0.76 | 1.13 | 0.78 | 0.18 | 0.35 | 0.63 | 0.39 |
| | 30% | **ActiveAD(Ours)** | **0.41** | **0.66** | **0.97** | **0.68** | **0.10** | **0.18** | **0.36** | **0.21** |

weather conditions. In line with previous works (Hu et al., 2022b;c; Jiang et al., 2023), we evaluate the planning performance using the Displacement Error (L2 loss) and Collision Rate metrics.

**End-to-end AD Models.** We selected latest works ST-P3 (Hu et al., 2022c), UniAD (Hu et al., 2022b) and VAD (Jiang et al., 2023) as our baseline models. Among them, the latest VAD demonstrates superior planning performance. Moreover, it achieves substantial reductions in computational overhead, and accelerates the training. Therefore, we adopt the lightweight version, VAD-Tiny, as the base model for subsequent experiments. We also include VAD-Based results in Appendix B.3.

**Active Learning Baselines.** As mentioned in Sec. 2.2, end-to-end autonomous driving is a novel and under-explored task for active learning. Directly transferring existing active learning methods, which are typically based on predictive probability analysis, is nontrivial. In particular, we select four methods as baselines that are relatively more transferable and relevant to this task: Coreset (Sener & Savarese, 2018): a feature selection-based approach; VAAL (Sinha et al., 2019): a task-agnostic method; KECOR (Luo et al., 2023a): a 3D Object Detection active learning method; ActiveFT (Xie et al., 2023b), which utilizes pre-trained features. Coreset utilizes the embeddings prior to the trajectory planning head (Jiang et al., 2023) as the input features. KECOR (Luo et al., 2023a) uses the public implementation to select proportional data for training. VAAL and ActiveFT take the raw images as inputs. The former employs an adversarial learning paradigm to discriminate unlabeled samples, while the latter uses ResNet50 (He et al., 2016) as the pretrained model for feature extraction, which is also adopted as the default backbone network in VAD (Jiang et al., 2023). ActiveFT selects all data within the budget at once, with no need for iterative selection.

**Implementation Details.** We set the annotation budget $B$ as 30% of the data volume: initially selecting 10% in the data pool, followed by an additional 10% in each subsequent selection round, for a total of two selection rounds. In each round, the model is retrained and used for the next round selection. We apply VAD-Tiny as the base model using the default hyper-parameter configuration. The confidence threshold $\epsilon_a$ and distance threshold $\delta_d$ is set to 0.5 and 3.0m respectively. For the initial selection, we set driving scenario threshold $\tau_c = 4$ and diversity partitioning parameter $\gamma = 0.5$. For the overall loss in Eq. 6, we normalize the criteria $\mathcal{L}_{DE}, \mathcal{L}_{SC}, \mathcal{L}_{AU}$ to $[0, 1]$ according to all scenes value respectively and set hyper-parameters $\alpha = 1$ and $\beta = 1$. We use AdamW (Loshchilov & Hutter, 2017) optimizer and Cosine Annealing (Loshchilov & Hutter, 2016) scheduler to train VAD-Tiny 20 epochs with weight decay of 0.01 and initial learning rate of $2 \times 10^{-4}$.

## 5.2 PERFORMANCE BY PLANNING METRICS

In Tab. 1 , we present the performance of all active learning models when choosing 10%, 20%, 30% of training samples. In Appendix B, we further give results of 40%, 50% and we observe that the performance is saturated at 30%, which again demonstrates the long-tail nature of AD data. We

Table 2: **Ablation for Designs.** "RA" and "ED" indicate the Random and Ego-Diversity based initial selection. "DE", "SC" and "AU" indicates Displacement Error, Soft Collision and Agent Uncertainty, respectively. All combinations with "ED" utilize the same 10% of data for initialization. The criteria $\mathcal{L}_{DE}, \mathcal{L}_{SC}, \mathcal{L}_{AU}$ are normalized to $[0,1]$ respectively and we set hyperparameters $\alpha$ and $\beta$ as 1.

| ID | Initiation | | Active Selection | | | Average L2 (m) | | | Average Collision (%) | | |
|----|----|----|----|----|----|----|----|----|----|----|----|
| | RA | ED | DE | SC | AU | 10% | 20% | 30% | 10% | 20% | 30% |
| 1 | ✓ | - | - | - | - | 0.86 | 0.82 | 0.78 | 0.67 | 0.54 | 0.37 |
| 2 | - | ✓ | - | - | - | 0.83 (−0.03) | 0.78 (−0.04) | 0.74 (−0.04) | 0.41 (−0.26) | 0.40 (−0.14) | 0.34 (−0.03) |
| 3 | - | ✓ | ✓ | - | - | 0.83 (−0.03) | **0.68** (−0.14) | 0.70 (−0.08) | 0.41 (−0.26) | 0.39 (−0.15) | 0.35 (−0.02) |
| 4 | - | ✓ | ✓ | ✓ | - | 0.83 (−0.03) | 0.81 (−0.01) | 0.73 (−0.05) | 0.41 (−0.26) | **0.35** (−0.19) | 0.26 (−0.11) |
| 5 | ✓ | - | ✓ | ✓ | ✓ | 0.86 (−0.00) | 0.80 (−0.02) | 0.71 (−0.07) | 0.67 (−0.00) | 0.38 (−0.16) | 0.26 (−0.11) |
| 6 | - | ✓ | ✓ | ✓ | ✓ | 0.83 (−0.03) | 0.76 (−0.06) | **0.68** (−0.10) | 0.41 (−0.26) | 0.39 (−0.15) | **0.21** (−0.16) |

Table 3: **Performance under Various Scenarios.** Average L2 (m) / Average Collision Rate (%) are reported under various weather / lighting and driving-command conditions, using 30% data selected by various active learning methods. The smaller value represents the better performance.

| Method | Weather / Lighting | | | | Driving-Command | | | | All |
|----|----|----|----|----|----|----|----|----|----|
| | Day | Night | Sunny | Rainy | Go Straight | Turn Left | Turn Right | Overtake | |
| Complete Data | 0.67 / 0.27 | 1.01 / 0.14 | 0.70 / 0.32 | 0.72 / 0.04 | 0.69 / 0.32 | 0.74 / 0.13 | 0.67 / 0.20 | 0.84 / 0.13 | 0.70 / 0.25 |
| Random | 0.72 / 0.26 | 1.29 / 1.25 | 0.78 / 0.39 | 0.79 / 0.26 | 0.70 / 0.22 | 0.89 / 1.03 | 0.86 / 0.32 | 1.05 / 0.22 | 0.78 / 0.37 |
| Coreset (Sener & Savarese, 2018) | 0.71 / 0.57 | 0.97 / 0.27 | 0.72 / 0.65 | 0.78 / 0.06 | 0.69 / 0.67 | 0.78 / 0.31 | 0.78 / 0.38 | 0.96 / 0.14 | 0.73 / 0.54 |
| VAAL (Sinha et al., 2019) | 0.78 / 0.34 | 1.09 / 0.34 | 0.80 / 0.40 | 0.89 / 0.12 | 0.79 / 0.38 | 0.86 / 0.34 | 0.82 / **0.20** | 0.96 / 0.18 | 0.81 / 0.35 |
| ActiveFT (Xie et al., 2023b) | 0.76 / 0.37 | 1.08 / 0.43 | 0.79 / 0.40 | 0.78 / 0.28 | 0.70 / 0.35 | 0.88 / 0.62 | 0.91 / **0.20** | 1.18 / 0.44 | 0.79 / 0.38 |
| **ActiveAD(Ours)** | **0.64 / 0.20** | **1.03 / 0.31** | **0.68 / 0.24** | **0.68 / 0.07** | **0.62 / 0.21** | **0.74 / 0.25** | **0.80 / 0.20** | **0.85 / 0.13** | **0.68 / 0.21** |

observe that traditional Active Learning methods perform poorly, lacking any significant advantage over random selection. In contrast, ActiveAD demonstrates significant advantages across the three different granularity ratios for data selection, highlighting the effectiveness of our method. This design enables improved sample selection and annotation for end-to-end planning-oriented autonomous driving. This is particularly relevant because manual annotation of samples for autonomous driving is resource-intensive and time-consuming. An astonishing finding is that **ActiveAD achieves comparable or even better performance by utilizing a carefully selected** 30% **of the data compared to training with the entire** 100% **dataset**. We believe that this finding is both intriguing and significant as it challenges the notion that more data necessarily leads to better performance. Current methods often focus on refining model structures while overlooking the importance of judicious data utilization. We argue that the data we select is more representative and informative, enabling to eliminate unnecessary noise and trivial samples that may cause adverse effects.

## 5.3 Ablation Study

**Effectiveness of Designs.** Tab. 2 shows the contributions of all the proposed components described in Sec. 4 to the final planning performance, including Displacement Error (L2) and Collision Rate. Our proposed Ego-Diversity based method exhibits superior performance in the initial 10% data selection, particularly in reducing the collision rate from 0.67% to 0.41%, thus providing a better initialization for subsequent model training.

During the subsequent active selection process, different metrics focus on various aspects. For instance, *Displacement Error* emphasizes the disparity between predicted and ground truth trajectories, effectively reducing the L2 loss of driving when used exclusively. However, the performance of Collision Rate remains unsatisfactory. Additionally, even with an increase in data volume, the results obtained using 30% of the data can be worse than those achieved with 20% of the data in terms of L2 performance. Indeed, focusing solely on a single metric can lead to overlooking other valuable information, potentially resulting in overfitting.

Moreover, we believe that avoiding collisions requires considering information from surrounding vehicles. Relying solely on Displacement Error makes it challenging to optimize the selection process. Therefore, the inclusion of the *Soft Collision* metric can improve performance in this aspect. When selecting 30% of the data, the collision rate decreased significantly from 0.35% to 0.26%, demonstrating a notable reduction. Additionally, considering the various possibilities of different objects in different environments, leveraging Agent Uncertainty can enhance the selection of complex scenarios. *Agent Uncertainty* assists in better optimizing both two planning metrics when the data volume increases. By incorporating these designs, ActiveAD has achieved outstanding performance. We also demonstrate that utilizing our incremental selection based on random initialization results in significant performance improvements, proving the effectiveness of each component individually.

Displacement Error | Soft Collision | Agent Uncertainty | Mixture (ActiveAD)

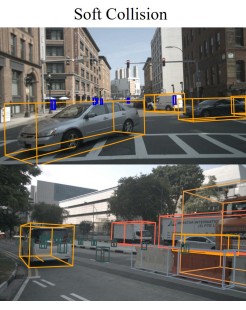
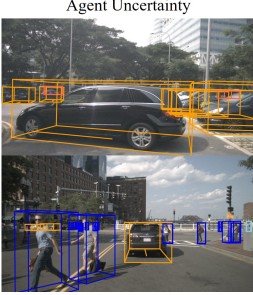
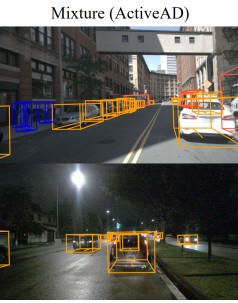

Figure 3: **Selected Scenes Visualization**. Front camera images selected according to the criterion of Displacement Error (col 1), Soft Collision (col 2), Agent Uncertainty (col 3) and Mixture (col 4) based on the model trained on $10\%$ data. 'Mixture' represents our final selection strategy ActiveAD, with considerations for the previous three scenarios.

**Ego-Diversity Hyperparameter Analysis.** We introduce the hyperparameter $\gamma$ in Sec. 4.1 to adjust the proportion of the initial selection based on the number of samples. In both real scenarios and model training, corner cases with fewer samples are often challenging and require special attention. Therefore, we choose to increase the focus on minority classes for $\gamma \leq 1$. Tab. 4 displays the results of our prelimi-

Table 4: **Ablation for Ego-Diversity Hyperparameter.** We enumerated the distributions obtained by selecting $10\%$ data under various $\gamma$ and compared their performance. # represents the numbers of scene occurrence.

| Diversity | Weather-Lighting | | | | Driving-Command | | | | Evaluation Metric | |
|---|---|---|---|---|---|---|---|---|---|---|
| Parameter | #DS | #DR | #NS | #NR | #S | #R | #L | #O | L2 (m) ↓ | CR (%) ↓ |
| Complete | 491 | 125 | 71 | 13 | 423 | 132 | 112 | 33 | 0.70 | 0.25 |
| $\gamma = 1$ | 49 | 12 | 7 | 2 | 40 | 13 | 11 | 6 | 0.90 | 0.46 |
| $\gamma = 0.8$ | 43 | 14 | 10 | 3 | 35 | 14 | 14 | 7 | 0.88 | **0.41** |
| $\gamma = 0.5$ | 34 | 17 | 13 | 6 | 27 | 17 | 16 | 10 | **0.83** | 0.43 |

nary experiments with different parameter values. We observe that when $\gamma = 1$, it ensures the stability of the selection process and provides velocity-based uniform selection compared to random selection. $\gamma = 0.8$ exhibits better performance in Collision Rate, while $\gamma = 0.5$ shows a clear advantage in Displacement Error (L2). Considering that the impact of Collision Rate diminishes when L2 is large, we select $\gamma = 0.5$ as the fixed parameter for subsequent model training and selection. Additionally, we did not extensively tune other parameters, such as $\alpha, \beta, \epsilon_a, \tau_c$, as their default values described in Sec. 5.1 already yielded satisfactory results.

**Various Scenarios Analysis.** We study the performance of active methods under diverse scenarios. Tab. 3 demonstrates that our method, ActiveAD, outperforms competitors in all cases, highlighting its superiority. ActiveAD exhibits strong robustness and excels in challenging situations, including rainy or nighttime conditions, as well as during overtaking maneuvers known for their higher difficulty. Furthermore, we achieve comparable performance using only $30\%$ of the available data, compared to utilizing the entire dataset.

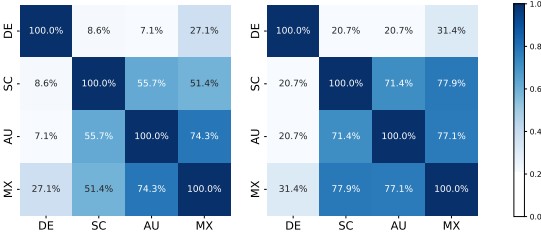

Figure 4: **Similarity between multiple criteria**. It shows the repetition rate of the $10\%$ (Left) and $20\%$ (Right) new sampled scenes selected by four criteria: Displacement Error (DE), Soft Collision (SC), Agent Uncertainty (AU) and Mixture (MX).

**Selected Scenes Visualization.** Based on the model trained on $10\%$ of the data, Fig. 3 illustrates the selection of representative scenarios using different metrics. The scenarios selected based on Displacement Error include complex maneuver trajectories such as lane changes and pedestrian avoidance. The scenarios selected based on Soft Collision often involve situations where the ego vehicle is in close proximity to other vehicles or obstacles, posing a risk. Examples include waiting at intersections for other vehicles to make turns, dense traffic in adjacent lanes, or situations with a high concentration of surrounding obstacles. Agent Uncertainty focuses on challenging road conditions, such as flickering lights, overtaking behaviors, vehicle reversing, and pedestrians crossing. ActiveAD combines considerations from all three criteria to select comprehensive samples across various scenarios. The top image shows an overtaking scenario, while the bottom image shows a nighttime following scenario. Fig. 4 illustrates the overlap rate among the

scenes selected based on these different criteria. In comparison, ActiveAD with mixture criterion demonstrates a better coverage of scenarios considered by individual criteria and emphasizes more on truly complex situations to enhance data quality for achieving excellent model performance.

## 5.4 Closed-Loop Evaluation

We implement ActiveAD in CARLA Town05 Short and Town05 Long similar to the protocol in ST-P3 (Hu et al., 2022c) and VAD (Jiang et al., 2023), and report the the Drive Score (DS) and Route Completion (RC) under different budgets as in Tab. 5. We could observe that the performance gets saturated after 20% with ActiveAD for the easier Town05 Short, possible due to the simpler driving logs in CARLA (simulation) than in nuScenes (real world). For Town05 Long, ActiveAD with 30% data achieves almost comparable performance compared to full data while random selection, the active learning baselines Coreset and ActiveFT perform consistently worse than ActiveAD, demonstrating the importance of planning-oriented data selection.

Table 5: **Closed-Loop Experiments in CARLA.** We chose 10% as the number of samples selected per round. ActiveAD achieves excellent results on both the Drive Score (DS) and Route Completion (RC) metrics. In contrast, the competitors do not demonstrate a significant advantage over Random.

| Method | Percent | Town05 Short | | Town05 Long | |
|---|---|---|---|---|---|
| | | DS↑ | RC ↑ | DS↑ | RC ↑ |
| Full | 100% | 63.11 | 88.90 | 33.24 | 76.33 |
| Random | 10% | 38.44 | 68.56 | 15.10 | 30.22 |
| ActiveFT (Xie et al., 2023b) | 10% | 39.15 | 68.01 | 15.22 | 31.48 |
| ActiveAD | 10% | 58.27 | 82.23 | 21.39 | 65.12 |
| Random | 20% | 47.46 | 77.20 | 20.01 | 55.99 |
| Coreset (Sener & Savarese, 2018) | 20% | 45.21 | 74.93 | 22.33 | 60.82 |
| ActiveFT (Xie et al., 2023b) | 20% | 48.63 | 78.71 | 21.37 | 58.22 |
| ActiveAD | 20% | 63.21 | 88.92 | 27.22 | 71.86 |
| Random | 30% | 55.81 | 81.04 | 23.25 | 60.44 |
| Coreset (Sener & Savarese, 2018) | 30% | 58.87 | 84.78 | 22.19 | 61.22 |
| ActiveFT (Xie et al., 2023b) | 30% | 56.54 | 82.11 | 24.62 | 63.34 |
| ActiveAD | 30% | 63.24 | 88.04 | 31.77 | 76.44 |

Notably, in the recently finished CVPR 2024 CARLA Leaderboard challenge, the winner solutions of both tracks (Renz et al., 2024; Jaeger & Chitta, 2024) use filtering techniques to reduce the number of simple training samples and achieve performance gains, which supports the claim that using all data for training in E2E-AD could be harmful for the planning performance.

## 6 Conclusion

In addressing the high cost and long-tail issues of data annotation for end-to-end autonomous driving, we are the first to develop a tailored active learning scheme ActiveAD. ActiveAD introduces novel task-specific diversity and uncertainty metrics based on a planning-oriented philosophy. Extensive experiments demonstrate the effectiveness of our approach, surpassing general peer methods by a significant margin and achieving comparable performance to the state-of-the-art model using only 30% of the data. This represents a meaningful exploration of end-to-end autonomous driving from a data-centric perspective, and we hope our work can inspire future research and discoveries.

**Limitations & Discussion.** In Appendix C, experiments demonstrate that model perception and prediction gradually strengthen with an increase in data volume. This is typical in similar fields, such as active learning for segmentation, and our method has not overcome this bottleneck. The first reason could be the inherent phenomenon in the 'predict-then-optimize' domain, where better predictions do not necessarily lead to better decisions. Thus, avoiding data redundancy and long-tail overfitting in E2E-AD becomes even more critical. Secondly, compared to the long-tailed distribution in planning, the repetition across different visual scenarios for vehicles can be relatively low.

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

## A  MORE DISCUSSION ABOUT ACTIVEAD

We believe that contributing to the community extends beyond proposing novel neural networks. **Identifying key issues and conducting preliminary explorations are equally vital. True innovation emerges from uncovering and understanding challenges, setting the stage for meaningful progress.** In this work, (1) We take the initial step to point out and analyze the data problem for E2E-AD. (2) Based on the characteristics of AD tasks, we design specific metrics to select samples which could optimize the planning performance by active learning, which fits planning-oriented spirits of E2E-AD. (3) The strong performance of the proposed method and the comprehensive ablation studies verify our claims.

What's more, we notice that recent events in the E2E-AD community further validate the major claims of our work:

1. In the recently finished CVPR 2024 CARLA challenge (June 2024) [1] , the winner solutions of both sensor and map tracks mention that they filter those less valuable frames during training. As a result, one winner state that by reducing the dataset size by 49%, with slightly improved performance. Their heuristic effectively removes redundant frames without losing information (Jaeger & Chitta, 2024).

2. Tesla, one of the world's leading autonomous driving technology company, claims that only about 1/10,000 of distance driven is useful for training, by their CEO Elon Musk in May, 2024[2].

We could observe that practioners in both academia and industry have both discovered the importance of data filtering. As the first work to study the data issue and active learning for E2E-AD, we believe the discoveies and insights of this work are worth sharing in the community.

## B  EXPERIMENTS DETAILS

### B.1  EXPERIMENTS SETUP

**End-to-end Autonomous Driving Models.** ST-P3 (Hu et al., 2022c) is an interpretable end-to-end vision-based network for autonomous driving that achieves better spatial-temporal feature learning. UniAD (Hu et al., 2022b) leverages information from multiple preceding tasks to enhance goal-oriented planning and demonstrates outstanding performance in all aspects, including perception, prediction, and planning. VAD (Jiang et al., 2023) introduces a vectorized paradigm as a substitute for the dense rasterized scene representation used in previous studies. This approach facilitates a more focused analysis of instance-level structural information, leading to excellent end-to-end planning performance. Moreover, it achieves substantial reductions in computational requirements, decreases the reliance on training devices, and accelerates training speed. Consequently, we adopt the lightweight version, VAD-Tiny, as the starting point for our experiments.

**Active Learning Baselines.** As mentioned in Sec. 2.2, end-to-end autonomous driving is a novel and underexplored task for active learning. It is difficult to directly transfer existing active learning approaches, which are usually based on predictive probability analysis, to this task. Therefore, we choose three classic methods that are more transferable and relevant as baselines: Coreset, a feature selection-based approach; VAAL, a task-agnostic method; KECOR (Luo et al., 2023a): a 3D Object Detection active learning method; ActiveFT, which utilizes pre-trained features. 1) Coreset (Sener & Savarese, 2018) formulates the data selection process as a k-Center problem on the learned embeddings of both labeled and unlabeled data. We utilize the features prior to the trajectory planning head (Jiang et al., 2023) as the embeddings. 2) VAAL (Sinha et al., 2019) employs the adversarial learning paradigm, utilizing a variational autoencoder (VAE)(Kingma & Welling, 2013) to extract image features from the nuscenes dataset, along with a discriminator network that distinguishes between labeled and unlabeled images. The VAE aims to deceive the discriminator by making it classify all samples as labeled data, while the discriminator strives to accurately identify the unlabeled samples in the data pool. Based on this approach, the selected unlabeled samples are then annotated. 3) KECOR (Luo et al., 2023a) identifies the most informative point clouds to acquire labels for 3D annotations through the lens of information theory. Samples selected based on this criterion are used for our end-to-end training. 4) ActiveFT (Xie et al., 2023b) uses pretrained features

---

[1]https://opendrivelab.com/challenge2024/#carla

[2]https://x.com/elonmusk/status/1787768103449010597

to optimize the distance between the distributions of labeled and unlabeled sets. In state-of-the-art autonomous driving methods, BEV features (Li et al., 2023a) are the commonly used representation. We adopt ActiveFT to use BEV features for data selection, and its strength lies in the ability to select all data under the budget at once, without the need for iterative selection.

**Annotation Budget**. In the scenario of active learning, the annotation budget is typically predetermined. Considering the complexity of end-to-end autonomous driving models and the diversity of tasks (including the final planning task as well as auxiliary perception and prediction tasks), we have set the annotation budget as 30%. Meanwhile, We further report the performance of ActiveAD with the budget from 10% to 50% of the data in Tab. 6. We observe that the planning performance is saturated around 30 % and thus we choose 30% as the stop threshold in the main paper.

Table 6: **All tasks' performance under different selection ratio.**

| Ratio | Planning | | Perception | | | | | | | Prediction | | | |
|---|---|---|---|---|---|---|---|---|---|---|---|---|---|
| | Avg. L2 ↓ | Avg. Col. ↓ | NDS ↑ | mAP ↑ | mATE ↓ | mASE ↓ | mAOE ↓ | mAVE ↓ | mAAE ↓ | minADE ↓ | minFDE ↓ | MR ↓ | EPA ↑ |
| 10% | 0.83 | 0.43 | 16.56 | 9.80 | 0.95 | 0.43 | 0.98 | 1.31 | 0.47 | 1.28 | 1.89 | 0.195 | 0.230 |
| 20% | 0.76 | 0.39 | 21.46 | 14.77 | 0.83 | 0.45 | 0.84 | 0.99 | 0.49 | 1.10 | 1.59 | 0.161 | 0.373 |
| 30% | 0.68 | **0.21** | 25.60 | 15.85 | 0.84 | 0.39 | 0.78 | 0.83 | 0.40 | 1.01 | 1.43 | 0.147 | 0.402 |
| 40% | **0.66** | 0.24 | 27.12 | 18.20 | 0.81 | 0.36 | 0.83 | 0.79 | 0.35 | 0.96 | 1.36 | 0.145 | 0.414 |
| 50% | 0.68 | 0.23 | 29.29 | 19.72 | 0.85 | 0.34 | 0.80 | 0.76 | 0.31 | 0.93 | 1.28 | 0.142 | 0.430 |
| 100% | 0.70 | 0.25 | **36.11** | **26.65** | **0.74** | **0.31** | **0.76** | **0.67** | **0.23** | **0.84** | **1.16** | **0.134** | **0.534** |

## B.2 METRICS EXPLANATION

In this paper, we utilize the evaluation metrics from VAD (Jiang et al., 2023), which is consistent with ST-P3 (Hu et al., 2022c). Therefore, the results from these two papers can be directly applied. Recently, inconsistencies in the UniAD metrics (Hu et al., 2023) have been identified within the community (Mao et al., 2023; Li et al., 2023b). We reference the content in (Mao et al., 2023) to provide more details about the evaluation metrics. The output trajectory $\tau$ is formatted as 6 waypoints in a 3-second horizon, i.e., $\tau = [(x_1, y_1), (x_2, y_2), ..., (x_6, y_6)]$. Then, the L2 loss is computed as:

$$l_2 = \sqrt{(\tau - \hat{\tau})^2} = \left[ \sqrt{(x_i - \hat{x}_i)^2 + (y_i - \hat{y}_i)^2} \right]_{i=1}^{6}, \tag{7}$$

where $l_2 \in \mathbb{R}^{6 \times 1}$ and $\hat{\tau}$ denotes ground truth trajectory. Then, the average L2 loss $\bar{l}_2 \in \mathbb{R}^{6 \times 1}$ can be computed by averaging $l_2$ for each sample in the test set.

UniAD (Hu et al., 2023) uses the value in the exact timestep as the L2 loss at the $k$-th second $(k = 1, 2, 3)$ :

$$L_{2,k}^{\text{UniAD}} = \bar{l}_2[2k]. \tag{8}$$

ST-P3 (Hu et al., 2022c) and VAD (Jiang et al., 2023) use the the average error from 0 to $k$ second as L2 loss at the $k$-th second:

$$L_{2,k}^{\text{VAD}} = \frac{\sum_{t=1}^{2k} \bar{l}_2[t]}{2k}. \tag{9}$$

Given the collision times $\mathcal{C} \in \mathbb{N}^{6 \times 1}$ at each timestep. Similarly, UniAD reports the collision $\mathcal{C}_k^{\text{uniad}}$ at the $k$-th second $(k = 1, 2, 3)$ as $\mathcal{C}[2k]$, while VAD reports $\mathcal{C}_k^{\text{VAD}}$ as the average from 0 to $k$ second.

Besides the variations in calculation methodologies, there is a distinction in the generation of ground truth occupancy maps between the two metrics. UniAD exclusively accounts for the vehicle category in creating ground truth occupancy maps, whereas ST-P3 and VAD incorporates both vehicle and pedestrian categories. This discrepancy results in different collision rates for the same planned trajectories when evaluated by these metrics, although it has no effect on the L2 error measurement. As a result, the collision rate in UniAD may be higher than reported, and this has been confirmed in (Li et al., 2023b) where VAD demonstrates superior performance in terms of collision rates. Consequently, we use a '-' in Tab. 1 instead of displaying specific values.

Taking into account the advantages of VAD in terms of model lightweighting (for instance, the ability to train using a 3090 GPU) as well as its leading position in comprehensive performance, we explore active learning based on the VAD model in this paper. This exploration is conducted from the perspective of data, aiming to provide insightful analysis.

Table 7: **Planning Performance with VAD-Base.** ActiveAD (w/o incremental) refers to the selection of all data solely based on diversity selection. ActiveAD (w/ incremental) indicates performing incremental selection based on an initial set.

| Base Model | Percent | Selection Method | Average L2 (m) ↓ | | | | Average Collision (%) ↓ | | | |
|---|---|---|---|---|---|---|---|---|---|---|
| | | | 1s | 2s | 3s | Avg. | 1s | 2s | 3s | Avg. |
| ST-P3 (Hu et al., 2022c) | 100% | - | 1.33 | 2.11 | 2.90 | 2.11 | 0.23 | 0.62 | 1.27 | 0.71 |
| UniAD† (Hu et al., 2023) | 100% | - | 0.42 | 0.64 | 0.91 | 0.67 | - | - | - | - |
| VAD-Base* (Jiang et al., 2023) | 100% | - | 0.39 | 0.66 | 1.01 | 0.69 | 0.08 | 0.16 | 0.37 | 0.20 |
| VAD-Tiny* (Jiang et al., 2023) | 100% | - | 0.38 | 0.68 | 1.04 | 0.70 | 0.15 | 0.22 | 0.39 | 0.25 |
| VAD-Base | 10% | Random | 0.49 | 0.81 | 1.20 | 0.83 | 0.38 | 0.57 | 0.91 | 0.62 |
| | 10% | ActiveAD(w/o incremental) | **0.48** | **0.76** | **1.14** | **0.79** | **0.24** | **0.43** | **0.68** | **0.45** |
| VAD-Base | 20% | Random | 0.47 | 0.78 | 1.15 | 0.80 | 0.32 | 0.47 | 0.75 | 0.51 |
| | 20% | ActiveAD(w/o incremental) | 0.44 | 0.75 | 1.10 | 0.76 | 0.25 | **0.34** | **0.61** | 0.40 |
| | 20% | ActiveAD(w/ incremental) | **0.42** | **0.70** | **1.08** | **0.73** | **0.16** | 0.35 | 0.64 | **0.38** |
| VAD-Base | 30% | Random | 0.44 | 0.74 | 1.08 | 0.75 | 0.16 | 0.34 | 0.54 | 0.35 |
| | 30% | ActiveAD(w/o incremental) | 0.42 | 0.71 | 1.05 | 0.73 | 0.14 | 0.29 | 0.49 | 0.31 |
| | 30% | ActiveAD(w/ incremental) | **0.40** | **0.67** | **0.93** | **0.67** | **0.09** | **0.21** | **0.35** | **0.22** |

Table 8: **Planning Performance with 5% Annotation Budget Per Selection Round on VAD-Tiny.**

| Selection Method | Percent | Average L2 (m) ↓ | | | | Average Collision (%) ↓ | | | |
|---|---|---|---|---|---|---|---|---|---|
| | | 1s | 2s | 3s | Avg. | 1s | 2s | 3s | Avg. |
| Random | 5% | 0.66 | 1.10 | 1.60 | 1.12 | 0.21 | 0.52 | 1.18 | 0.64 |
| ActiveAD | 5% | **0.63** | **1.04** | **1.51** | **1.06** | **0.15** | **0.49** | **1.02** | **0.55** |
| Random | 10% | 0.51 | 0.83 | 1.23 | 0.86 | 0.40 | 0.62 | 0.98 | 0.67 |
| ActiveAD | 10% | **0.45** | **0.81** | **1.17** | **0.81** | **0.17** | **0.38** | **0.76** | **0.44** |
| Random | 15% | 0.49 | 0.81 | 1.21 | 0.84 | 0.27 | 0.54 | 0.84 | 0.55 |
| ActiveAD | 15% | **0.47** | **0.76** | **1.15** | **0.79** | **0.24** | **0.37** | **0.63** | **0.41** |
| Random | 20% | 0.49 | 0.80 | 1.17 | 0.82 | 0.36 | 0.49 | 0.77 | 0.54 |
| ActiveAD | 20% | **0.43** | **0.77** | **1.11** | **0.77** | **0.19** | **0.35** | **0.66** | **0.40** |
| Random | 25% | 0.47 | 0.77 | 1.13 | 0.79 | 0.23 | 0.37 | 0.59 | 0.40 |
| ActiveAD | 25% | **0.41** | **0.69** | **1.05** | **0.72** | **0.16** | **0.29** | **0.54** | **0.33** |
| Random | 30% | 0.45 | 0.76 | 1.12 | 0.78 | 0.17 | 0.30 | 0.63 | 0.37 |
| ActiveAD | 30% | **0.42** | **0.67** | **1.00** | **0.70** | **0.08** | **0.19** | **0.41** | **0.23** |

## B.3 EXPERIMENT RESULTS FOR VAD-BASE

Tab. 7 presents the experimental results of our method based on the VAD-Base model. Compared to the baseline of random selection, our method—whether it be the one-time sample selection based on Ego-Diversity or the complete method that performs Incremental Selection starting from an initial dataset—has shown significant advantages. Consistent with the conclusions in the main paper, using 30% of the data, our approach achieves performance on par with using the entire dataset, validating the effectiveness and universality of our method.

## B.4 EXPERIMENT RESULTS FOR DIFFERENT ANNOTATION INCREMENT.

In active learning, the sample selection ratio in each round plays a crucial role in determining performance. In our main experiments, we chose 10% as the number of samples selected per round. Here, we set the interval to 5%, resulting in training data proportions of 5%, 10%, 15%, ..., up to 30%. Since other active learning methods do not show a significant advantage over random selection, we present a comparison of our method ActiveAD, with Random in the Tab. 8. Our findings reveal that, across different initialization ratios and selection intervals, ActiveAD consistently demonstrates robust performance advantages, underscoring its versatility across various labeling scenarios.

## C   PERCEPTION AND PREDICTION PERFORMANCE.

Existing end-to-end training models (Hu et al., 2022b; Jiang et al., 2023) often utilize visual information as auxiliary tasks to assist core objective planning. The main experiment shown in Tab. 1, demonstrates our advantage in planning metrics, while we are also curious about perception and prediction task performance. Tab. 6 displays the performance after training with different proportions of data. The perception metrics include NDS(nuScenes detection score), mAP(mean Average Precision), mATE(mean Average Translation Error), mASE(mean Average Scale Error), mAOE(mean Average Orientation Error), mAVE(mean Average Velocity Error), mAAE(mean Average Attribute Error) which are sourced from the nuScenes dataset setting (Caesar et al., 2020). The prediction metrics include minADE (minimum Average Displacement Error), minFDE (minimum Final Displacement Error) and MR (Miss Rate) and EPA (End-to-end Prediction Accuracy) (Hu et al., 2023).

It can be clearly observed that there still exists a significant performance gap in these metrics between utilizing a small amount of data and using complete data. This observation aligns with common sense in active learning tasks (Sener & Savarese, 2018; Sinha et al., 2019; Xie et al., 2023b; Zhan et al., 2022), where a small sample size can not outperform the entire dataset in traditional image classification and segmentation tasks. We would like to offer some thoughts on this phenomenon.

- In the field of optimization with uncertain coefficients, recent studies (Elmachtoub & Grigas, 2022; Cameron et al., 2022; Mandi et al., 2020) have also found that when a task involves both prediction and decision-making, with the prediction output serving as the input for the decision task, there can be a misalignment between the optimization objective of the prediction and the overall decision objective. In other words, better predictions do not necessarily lead to better decisions. For example, in the shortest path problem on a graph with unknown paths mentioned in Elmachtoub & Grigas (2022), when the path costs are predicted using a dataset during the prediction phase and directly used for downstream solving, the solution obtained is not optimal.

- The enhanced visual perception capabilities afforded by larger datasets can be expected and align with common sense within the Active Learning community. A plausible explanation is that while driving trajectories might show a long-tail distribution, the repetition across different visual scenarios for vehicles is relatively low. Different environments, road sections, and lighting conditions inevitably lead to varied scenarios, making saturated training valuable. However, in end-to-end AD tasks, where these serve as auxiliary losses, our primary goal is decision-making, specifically trajectory planning. Thus, avoiding data redundancy and long-tail overfitting becomes even more critical.

It also raises the question of how to balance other losses in E2E-AD, considering planning as the ultimate objective, and whether there are better training paradigms. Our active learning approach provides a means to optimize training data while reducing costs. We believe that future work on multitask learning or hard case mining holds promise for enhancing planning performance.

