# OpenReview forum: "ActiveAD: Planning-Oriented Active Learning for End-to-End Autonomous Driving"
_ICLR.cc/2025/Conference — ICLR 2025 Conference Withdrawn Submission_

### Official Review · Reviewer_pMRE · 2024-10-30

**Soundness:** 3
**Presentation:** 4
**Contribution:** 3
**Rating:** 6
**Confidence:** 4

**Summary:**

The paper proposes an active learning strategy for selecting which part of the dataset is useful
for annotation in order to improve the model.
They use modular end-to-end driving as a way to incrementally discover which parts of the data to annotate.
They obtain very compelling results in both offline and closed loop evaluation in CARLA simulator.

**Strengths:**

The main strong point is the relevance of the problem. Data annotation is
the main bottleneck for most machine learning applications, autonomous driving included.
The paper shows that it is possible to get comparable results using only 30% of the data.
Selecting this data properly is of very high relevance and we should see more papers
like this.

**Weaknesses:**

The contribution is generally simple: Use the motion prediction formulation for
active learning. If this is some first results on this matter, I believe the contribution is relevant.

On the results section, the fact that active learning using motion prediction is superior
 than using other downstream tasks is not particularly impressive. I think the
ablation section showed more insights on that matter. This made me miss
the ablation for CARLA results which are definitely more conclusive than a single
dataset open loop evaluation.


I know resources is a big issue on running experiments for this type of domain but I am
strongly inclined to believe that this method has a high variability if you retrain different random
seeds since it involves several training processes. The position where you stop the training
might give a big variation. I wonder if the results obtained would resist if another random seed
was trained and if the data selected would remain consistent.

**Questions:**

Having a way to benchmark the data selected is key here, my view.

I am particularly interested in having some more insight on the scenarios selected.
Figure 3 shows some of those scenarios with respect to the different metrics but I would be interested
in a general distribution of scenario types selected when using the automatic selection
versus when not using it.

---

### Official Review · Reviewer_A5zN · 2024-11-02

**Soundness:** 2
**Presentation:** 2
**Contribution:** 2
**Rating:** 3
**Confidence:** 5

**Summary:**

This work explores active learning for end-to-end autonomous driving. In particular, the authors design the Ego-Diversity metric for initial selection. Then, three criteria, namely Displacement Error (DE), Soft Collision (SC), and Agent Uncertainty (AU), are introduced for incremental selection. VAD is used as the baseline method, on which the authors prove the effectiveness of the proposed approach.

**Strengths:**

- Selecting critical data for end-to-end tasks in autonomous driving is essential.
- Utilizing annotation-free trajectories to develop the strategy is a prudent choice.
- Experiments have demonstrated the effectiveness of the self-conducted baseline.
- The writing is clear and easy to follow.

**Weaknesses:**

Every coin has two sides, and the strengths I mentioned are no exception.

- Data selection using AI models sounds appealing; however, a frequently updated selection model is impractical for autonomous driving. It is well known that data in real-world autonomous driving systems is updated daily, which can introduce distribution shifts from previously collected data. This necessitates retraining the active model with each new data influx, which is not resource-efficient.

- While using trajectories is direct and simple, end-to-end driving systems are designed to fully leverage scene information. Relying solely on trajectories may make the proposed method more suitable for classical motion planning rather than end-to-end driving.

- The comparative analysis lacks consideration of recent works, and it is not fair to compare this approach with other active learning methods primarily designed for visual intelligence.

**Questions:**

Please compare it with more recently proposed methods.

---

### Official Review · Reviewer_QZzR · 2024-11-04

**Soundness:** 2
**Presentation:** 2
**Contribution:** 2
**Rating:** 3
**Confidence:** 5

**Summary:**

The authors propose an active-learning framework for data selection in end-to-end autonomous driving.
The framework contains an initial data selection stage and an incremental data selection stage.
The authors leverage diversity-based metrics(e.g. weather, light, driving commands and average speed) for the initial data selection stage.
For the incremental selection stage, planning performance metrics like trajectory ADE, collision scores as well as the uncertainty of other road user’s future trajectory prediction are used for selecting incremental training samples.
Experiments on open-loop nuScenes show marginal improvement while experiments on CARLA shows more significant improvement.

**Strengths:**

1. The problem studied is important. Active learning is a crucial problem for autonomous driving, especially for end-to-end planner where generally a large amount of data is used for training and data curation is a key step.
2. This work is good early attempt at using driving specific metrics instead of general classification metrics for active learning in autonomous driving.

**Weaknesses:**

1. industry as well reported in CVPR competitions and the recent NAVSIM work. I could easily come up with a lot of other metrics like the road type(highway, urban, rural), road topology(intersection, T-junction, U-turn, etc), traffic density(how many road users in the scene) and sunlight angle(front, back, side). So I don’t quite understand what the scientific challenge the authors trying to solve here or just try to report a proactioner’s guide? The incremental selection stage mostly uses the evaluation metrics of the planner and the predictor, which is a very general approach of adding more data at what the model is bad at. Overall, I think this work is more of a technical report instead of a research paper.
2. The baselines are way too weak. Most baselines except KECOR are designed and evaluated for the image classification task. I don’t think these method could transfer at all for the end-to-end driving. The authors should come up with reasonable but simple baselines and highlight the technical challenges in the problem instead of running experiments on a set of so-called baselines that are known not work and even underperform the random baseline.

**Questions:**

N/A

---

### Official Review · Reviewer_isaw · 2024-11-05

**Soundness:** 2
**Presentation:** 3
**Contribution:** 2
**Rating:** 3
**Confidence:** 4

**Summary:**

The paper proposes ActiveAD, an active learning framework for end-to-end autonomous driving. One major challenge in E2E-AD lies in the expensive data annotation process and the long-tailed distribution of AD datasets, where much of the collected data is redundant (e.g., straightforward driving on empty roads). ActiveAD addresses these issues by designing the following metrics. Ego-Diversity: A diversity-based initialization method that considers weather, lighting, and driving behaviors to address cold-start issues. Planning-Oriented Metrics: The use of Displacement Error, Soft Collision, and Agent Uncertainty metrics for iterative sample selection to reduce the annotation burden while maintaining high planning performance. The paper demonstrates data-efficiency improvements by achieving state-of-the-art performance using only 30% of training data on both the nuScenes dataset and CARLA simulation.

**Strengths:**

1. This paper studies the problem of active learning in self-driving. It is an important problem for scalable developments and faster iterations for industry. The focus on planning performance for active learning is new. Existing active learning methods mainly optimize perception / prediction tasks, but ActiveAD extends this to planning in E2E-AD.

2. The paper is well written and easy to follow. The metrics proposed in this paper are intuitive and straightforward. The combination of displacement error, soft collision, and agent uncertainty provides a robust way to identify critical data samples for annotation. I also like thorough ablation studies presented in the paper.

**Weaknesses:**

1. Insufficient evaluation and limited generalization to real-world scenarios. It is particularly important for this paper to demonstrate the proposed metrics can be adapted to different datasets and architectures rather than just some heuristics-based tuning on specific datasets. It is not a big surprise that using 30% data can achieve on-par performance with careful data selection. Moreover, this paper does not evaluate the robustness of the trained autonomy on more extreme / out-of-distribution (OOD) settings (e.g., safety-critical scenarios, extreme weather, complex interactions, etc). I do not believe the metrics on the eval set can tell the full story. In general, my biggest concern is how generic the proposed metrics are and how robust the trained model is. Current evaluation on nuScenes and CARLA is not sufficient. I would recommend testing on larger datasets (e.g., argoverse, waymo) and more diverse e2e models.

2. This paper misses a significant amount of work for active learning in the self-driving domain. For instance, the seminal work is not discussed in the paper. There are also a lot of follow up works (e.g., [2][3]). More comparisons and discussions with them would be beneficial. Another baseline to consider is getting some planning costs for each scene and pick the hardest ones.

[1] Scalable Active Learning for Object Detection. Haussmann et al., 2020. \
[2] Just Label What You Need: Fine-Grained Active Selection for Perception and Prediction through Partially Labeled Scenes. Segal et al., 2021. \
[3]  Improving the Intra-class Long-Tail in 3D Detection via Rare Example Mining. Jiang et al., 2022.

3. There are too many parameters to tune and the procedure is quite complicated. How can we make sure we choose the correct setting when in the production setup (say we need to train a new model based on newly collected data and we cannot tune). I am a bit worried about the real impact of the proposed paradigm as there is no automatic data selection procedure involved like many other works (e.g., using the training loss, entropy in the prediction etc). Also, it seems that the planning improvement is quite limited when training more data (30% -> more data) but perception and prediction can continue improve from Table 6. The mAP with 30% data is only 15.85 vs 26.65 which is a signficant performance drop. I am worried that using this paradigm will give us less robust models (the planning metrics can be noisy and cannot tell the full story?).

4. There are a lot of places where the bold highlights are wrong. For instance, in Table 3, on night scenario, coreset results 0.97 / 0.27 is actually better than the ActiveAD. In rainy and turn right, coreset results 0.06, 0.78 are better ActiveAD. In Table 1, VAD-Tiny 20% data, VAAL average collision error for 1s is smaller. I recommend the authors to carefully check the tables.

**Questions:**

N/A

---

### Note · Authors · 2024-11-13

**Comment:**

Thank all the reviewers for their valuable time and comments. We will make further revisions to the paper based on the constructive and insightful suggestions. Thank you once again.

**Withdrawal Confirmation:**

I have read and agree with the venue's withdrawal policy on behalf of myself and my co-authors.